# Managing Reputation in MNEs through Intangible Liabilities

**Maria-José García-López** [1,*], **Juan-José Durán** [2] **and Carmen Avilés-Palacios** [3,*]

1 Department of Economía Financiera y Contabilidad, Universidad Rey Juan Carlos, 28032 Madrid, Spain
2 Department of Financiación e Investigación Comercial, Universidad Autónoma de Madrid, 28049 Madrid, Spain; juanjose.duran@uam.es
3 Department of Ingeniería y Gestión Forestal y Ambiental, Universidad Politécnica de Madrid, 28040 Madrid, Spain
* Correspondence: mariajose.garcia@urjc.es (M.-J.G.-L.); carmen.aviles@upm.es (C.A.-P.); Tel.: +34-914-887561 (M.-J.G.-L.); +34-910-671632 (C.A.-P.)

**Abstract:** Company-specific assets, especially those of an intangible nature, are sources of value for the firm; consequently, the company should carry out a good management of them to increase the firm's competitiveness, accesses to financing, reduce risks and improve its reputation. However, no attention has been paid to the concept of intangible liabilities that a company may create or develop and its effects on the value of the firm, and the relationship with its competitiveness and reputation, with reputation being considered as the public recognition (perception) of the quality of activities of the firm by both internal and external stakeholders of the organization. The right identification of them should allow us to better manage companies. Through literature analysis and empirical observations, we identify different sources of intangible liabilities and their effect on the value of the firm, highlighting those of them that implies a negative impact in the firm reputation. We identified which factors are greatly impacted when firm reputation decreases, so that a constructor is proposed to explain the relationship among reputation and the potential of generating implicit intangible liabilities. As result of an empirical analysis, we conclude that the factors (corporate assets, quality of management, financial soundness and long-term investment) impacting more on perception of reputation by stakeholders are those linked to the management style of the MNE. This would help MNCs to better manage both intangible assets and liabilities.

**Keywords:** intangible liabilities; intangible assets; firm reputation; multinational firms

## 1. Introduction

Multinational firms (MNE), by definition, have relations with home and local stakeholders residing in the countries where they operate. At the same time, there are global stakeholders, with an interest in the whole and in the parts of MNEs, such as international organizations, international activist groups (NGOs), and institutional investors, other multinationals and internationalized firms. In this sense, there are also overlapping stakeholders' groups in different countries [1]. Stakeholders in one country may also decide to take part on behalf of stakeholders in other countries when faced with certain events or facts related to corporate responsibility (child labor, environmental issues, etc.) [2].

Reputation is created or build overtime and it might be a long-lasting perception, often difficult to change [3,4]. For example, a reputation may be maintained even after product quality has dropped (for example, Volkswagen) or that competitiveness has not been updated.

The company's voluntary behavior responding to social and environmental concerns, develops intangible assets which can be sources of competitive advantage [2]. When causes of bad reputation arise and using the same argument, irresponsible behavior brings about intangible liabilities (both for the company and for society).

From our perspective, the identification of intangible liabilities affects negatively firm reputation. Consequently, the possibility to arise intangible liabilities should be more

carefully controlled on high reputation firms. From this point of view, we define intangible liabilities as those amounts, neither insured nor hedged, that the company is required to recognize as a contingency or as debt, at the time having the knowledge that an event has occurred, that will force the company to disburse economic quantities in the future, whatever the origin. In this sense, we can mention various sources of intangible liabilities such as mismanagement, deliberate concealment of facts, and no anticipated negative external events, such as catastrophes. Usually, the registered debt in a firm's accounts is what we can name explicit contractual debt. The amount, maturity and interest rate of the debt are explicitly written in the contract. However, a firm may generate another debt: implicit potential debts as well as created reserve funds to hedge contingencies linked to its operations and activities.

The first article dealing with the concept of intangible liabilities was written by Harvey and Lusch and published in the European Management Journal [5]. They considered intangible liabilities as hidden value of obligations which will imply payments to other entities in the future. A decade later, Garcia-Parra et al., from an accounting perspective, reviewed all literature available on this topic and exposed that, excepting the cited contribution, intangible liabilities were treated as depreciation of intangible assets by the few authors (six) who had written about it [6]. They defined intangible liabilities as the organisational non-monetary obligations that the company must accept and acknowledge to avoid the depreciation of its intangible assets, arguing that intangible liabilities are the consequences of a reduction in intangible assets due to tacit knowledge, that take the form of obligations. However, after meditating on this reading, we have not found a clear explanation in this paper of the relationship between the two variables. The literature considers intangible assets, but intangible liabilities are not studied as much [7].

The main objective of this paper is defining intangible liabilities as a new concept not deeply studied in business economics nor in international business literature. Secondly, we identify the sources of this intangible liability that it may be related to the competitiveness of the firm, considering that high reputation firms should have minimum intangible liabilities to maintain their high reputation. We also argue the correlation expected between the competitiveness of the firm and the specific and singular intangible asset such as *reputation* to analyse which factors are more greatly perceived by stakeholders as a source of intangible liabilities in MNE and the relationship to the firm's reputation, being the main consequence of this.

The state of the art regarding intangible assets and liabilities and its relationship with reputation is stablished throughout the analysis of the, not very abundant, literature and cases studies [7]. With this base, an empirical study is run to find the statistical relationship among attributes of reputation and financial figures, defining some constructors and studying the correlation between reputation and financial performance to reach the conclusions.

The paper is organized as follows. In Sections 2 and 3, the hypothesis and materials and methods followed will be explained. Sections 4 and 5 offer a literature approach to intangible assets and intangible liabilities seeing their correlation; regarding reputation as a valuable intangible asset, we analyze the different sources of intangible liabilities supported by real examples. The results and discussion are explained in Section 6, where reputation as a constructor indicates the potential of generating implicit intangible liabilities. Finally, a conclusion section focuses on reputation relationship with quality and style of management, long term investment and financial soundness and not only to financial performance, and offers the limits of this study, and future research.

Our analysis shows that the factors (corporate assets, quality of management, financial soundness, and long-term investment) with a high impact on the stakeholder's perception of reputation are those linked to the management style of the MNE. The remaining factors (innovation, human resources management, corporate social responsibility, quality of products and/or services and global competitiveness) are consequences of management decisions.

## 2. Objectives and Hypothesis

The objectives of this paper are (1) defining intangible liabilities; (2) identify sources of this intangible liability and its relationship with competitiveness of the firm and its reputation. Therefore, the hypotheses beneath our research are the following:

**Hypothesis 1 (H1):** *Factors impacting more on perception of reputation by stakeholders are those linked to the management style of the MNE.*

**Hypothesis 2 (H2):** *Remaining factors such as innovation, human resources management, corporate social responsibility, quality of products and/or services and global competitiveness are consequence of management decisions.*

**Hypothesis 3 (H3):** *Factors such as innovation and quality of products and services are relevant in the study of reputation variation.*

## 3. Material and Methods

To contrast the hypotheses, we considered a twostep methodology. Firstly, we studied the actual research regarding intangible assets and liabilities and its relationship with reputation; we explained sources of intangible liabilities using some relevant study cases. Secondly, we carried out an empirical study to find the statistical relationship among attributes of reputation and financial figures.

As the main purpose of our study is to analyse whether a firm's reputation is more affected by the defined key attributes or by a firm's financial performance, we generated two constructors to study which factors are affecting the most each variable. Once we have created these constructors, we will study the correlation between reputation and financial performance to reach the conclusions.

In this study, we selected the top 50 MNE companies in the index "Fortune, the Most Admired Companies in the world" in 2017, 2018 and 2019 [8–10]. This ranking is chosen because it analyses companies' reputation. The attributes of reputation on which companies are evaluated in determining the industry rankings, are: 1. Ability to attract and retain talented people; 2. Quality of management; 3. Social responsibility to the community and the environment; 4. Innovativeness; 5. Quality of products or services; 6. Wise use of corporate assets; 7. Financial soundness; 8. Long-term investment value; 9. Effectiveness in doing business globally. We have focused our research on these years using the schedule for the European Commission sustainable finance taxonomy working group, created in 20218, which published a first draft in 2019. The aim of this taxonomy is to consider sustainable activities as a discriminator for financing. In our opinion, an increase in the commitment for sustainability would produce a reduction in potential risks, a better access for financing, and in the reputation of a company as well [11]. Results would offer a baseline for future research when taxonomy will be widely implemented.

These attributes were developed prior to the inception of the Most Admired Companies rankings in the mid-1980s through a series of interviews with executives and industry analysts to determine the qualities that make a company worthy of admiration. Only the attribute names as listed above are provided on the survey. They simply state that ratings may be based on the respondents' first-hand knowledge of these companies or on anything they may have observed, heard, or perceived about them. Thus, interpretation of the meaning of the attributes within a specific industry is left to the respondents. For that reason, we conclude that the variation in reputation is the variable to study since determination is based on people's perception of the emergence of intangible liabilities that decrease a firm's reputation.

We have also considered the main financial figures relating to these companies defined by market capitalization, net profit, sales, EBITDA (Earnings before interest, taxes, depreciation, and amortisation), price-to-book value, total assets, total tangible assets, total

debt and total equity. Theses variables have been considered useful for this kind of study in similar works [11–19].

The main statistics to validate internal consistency and reliability are also included in these two significant constructors: (1) The first one was composed by the use of corporate assets, quality of management, financial soundness and long-term investment value; (2) The second one was composed by Innovation and Quality of Products and Services.

In the same sense, the other variables were not related, following the factor analysis, on the explanation of reputation variation. Afterwards, factor analysis of all selected financial variables was performed, and the result gives us two significant constructors.

- The first financial synthetic variable is constructed by the factor relationship among assets, liabilities and shareholders' equity, coming from the basic accounting equation and the relationship between economic resources and financing sources and following the statistical demonstration of Stowe et al. [11].
- The second financial synthetic variable is constructed through the factor relationship of market capitalization, revenue, book value and tangible assets, as it is pointed out by Ryan [12].

A factor analysis was conducted to determine which of the variables compose each synthetic variable. For an item to be accepted in this exploratory study (EFA), it must meet a minimum *factor loading* of 0.7. Next, the internal consistency and reliability of each of the resulting dimensions are analyzed using the Cronbach's alpha coefficient and the average variance extracted (AVE).

Once the constructors are defined, the relationships between them are analyzed. Using SPSS software, all the variables that compose a synthetic variable were transformed into one single factor to analyze the correlations between them. Finally, we study the correlation among all the factors included in each synthetic variable to contrast the results we obtained previously.

Using this methodology offers a wide approach to the relationship among intangible liabilities, management and reputation under a statistical point of view and a baseline to a replicate study using near in time data. This further research will offer the impact of sustainable policy in MNEs management and performance

## 4. Intangible Assets and Liabilities in Multinational Business Research

Explaining intangible liabilities need to start from defining what intangible assets are, at least under an accounting point of view they are related.

Two main sources of intangibles (we only cite some of the main representative authors on these topics) have been considered as relevant determinants factors of foreign direct investment (FDI) decisions and Multinational Enterprises (MNE) strategies. On one hand, we mention the Liabilities of Foreignness associated with operating in a foreign country based on managing differences (distances) between countries of origin and country of destination of FDI, that brings cultural, and physic distance. It includes six dimensions: (1) education, (2) industrial development, (3) language, (4) degree of democracy, (5) political, ideology and religion, and (6) institutional distance as research agendas in International Business (IB) [16,20–23]. On the other hand, we face intangible assets (ownership advantages) based on knowledge as the main sources of the economic capital of MNEs and its competitive advantages [24–26].

Thinking on these two types of intangibilities we wonder: are there any type of intangible liabilities related to MNEs activities? If so, are we missing an important concept and research area in international business? To answer these questions, we make the following considerations. A MNE is competitive if is able and knows how to produce, commercialize, distribute, and sell goods and services in the market either cheaper or differentiated by nature and quality characteristics from those made by competitors (selling to the same market target). The competitiveness is based on firm´s specific knowledge and capabilities that are incorporated into physical and intangible assets. These types of assets together with another of a generic type, constitute what can be called Economic

Capital of the firm. By nature, those assets can be of a technological character (that allow to manufacture or produce goods and services), commercial (for distribution and selling) and managerial (design and implementing strategies, assign resources, coordinate, and control). Physical specific assets can be found in all the three types of economic capital we have mentioned. However, nowadays it is possible to consider, due to its singularity, another kind of intangible asset: The reputation of the firm.

In our understanding, we believe that reputation can be conceptualized as an intangible asset. Reputation is based on public recognition or perception of the quality of activities of the firm, by both internal and external stakeholders of the organization [27]. However, even if perceptions logically differ between individual stakeholders, it could be understood as a collective perception [28–30]. Favorable collective perception may be a source of economic rents, and then there are incentives for firms to maintain and invest in their reputations [31,32].

The relationship between reputation and a sustained competitive advantage is widely acknowledged in the literature [33,34]. Some researchers have found a link between reputation and organizational performance [33,35]. Reputation is the single most valued organizational asset and as such can be taken as one of the most relevant strategic resources [36–38]. Even reputation, understood as having a reliable, fair, and honest behaviour, can be advantageous for MNE in the eyes of its stakeholders as well as for their internationalization strategies [38]. Reputation is a signal of quality management and relevant factor to deal with uncertainty asymmetry [39,40].

The context in which signalling occurs matters [38]. The underplaying causes or determinants of reputation are, to a great extent, firm-specific and path dependent [38]. Reputation is also important in emerging markets due to higher cost in building credible brands and not damage brand quality [38].

Reputation contributes to generate goodwill of the firm and negative reputation may contribute to create bad-will and may leads to generate intangible liabilities, as well. A positive reputation increases the likelihood that stakeholders will be favourable to contract with a recognized firm [41,42]. Within this argument, not behaving reliably or honestly can have immediate and long-term consequences, as a decrease in positive reputation may affect the future actions of other players toward a firm. If the "present value of future income exceeds the short-term profit" of dishonesty, firms will be honest and invest in their reputations.

Therefore, the efficient management of this economic capital increases the value of the firm and its reputation. A firm responsible with its stakeholders will also fulfill its reputation. In this sense, stakeholders' knowledge of intangible and recognized liabilities should not affect the firm' reputation. The competitive advantages of the firm can be codified (documents, stored, structured, and can be registered as patents, marks, trade names, logos, industrial design, etc.) or may not be able to be recorded, this being known as tacit knowledge. The former is explicit and easily transferred or commercialized, whereas the latter are implicit and not transferable outside the organization that owns them. Tacit assets basically stored in the human mind are generated through individual and team experience. This implicit knowledge only can be acquired through merger and acquisition and with effective integration within the new organization.

The assets of a firm, specific and generic, in general are valued at historical cost and are incorporated in the balance sheet. (The firm can estimate the replacement value of assets as proxy of its current prices and compare it to the market value of the firm (equity plus debt) and observe if there exist intangible assets (and liabilities) not incorporated in the accounts of the firm). Generally, only those intangible assets acquired (patents, trademark, goodwill) in the market as well as investments made in R&D, are included in the asset side of the balance sheet. Can a negative goodwill (badwill) be understood as an intangible liability already paid by the firm´s equity owners?

When there is a loss of reputation, companies make large investments to recover it, what probably increases risks and financial needs. Therefore, the emergence of intangi-

ble liabilities has economic consequences that directly affect the value of the company. Therefore, it is possible to assume that a firm may developed intangible liabilities that may imply monetary payments to third parties either at a specific time or as a future cash outflow because of applying bad practices, following wrong policies on a continuous base or due to events as results of concrete activities. Additionally, no anticipated negative events or hidden implicit contracts may result in potential intangible liabilities that, in the future, may needed to become explicit. When the intangible liability is related to bad reputation and less competitiveness it has also consequences in a reduction in the equity value for shareholders.

Intangible assets and liabilities are two faces of the same coin, intangible liabilities are opposed to intangible assets as they produce losses that are difficult to quantify. On the other hand, they generate strategic value and may be accounted [35]. Figure 1 shows the accounting equation balance between assets and financing sources. In our approach, a decrease in intangible assets implies a loss in equity and, in some cases, the emergence of intangible liabilities. Intangible liabilities may therefore affect both partners and third parties: partners because the decrease in intangible assets value, and third parties because intangible liabilities can arise in form of fines, contingencies, payments to be declared, that affect stakeholders of the firm. Following this argument, intangible assets increase the value of the firm, and intangible liabilities may decrease the value of equity, in certain cases losing competitiveness.

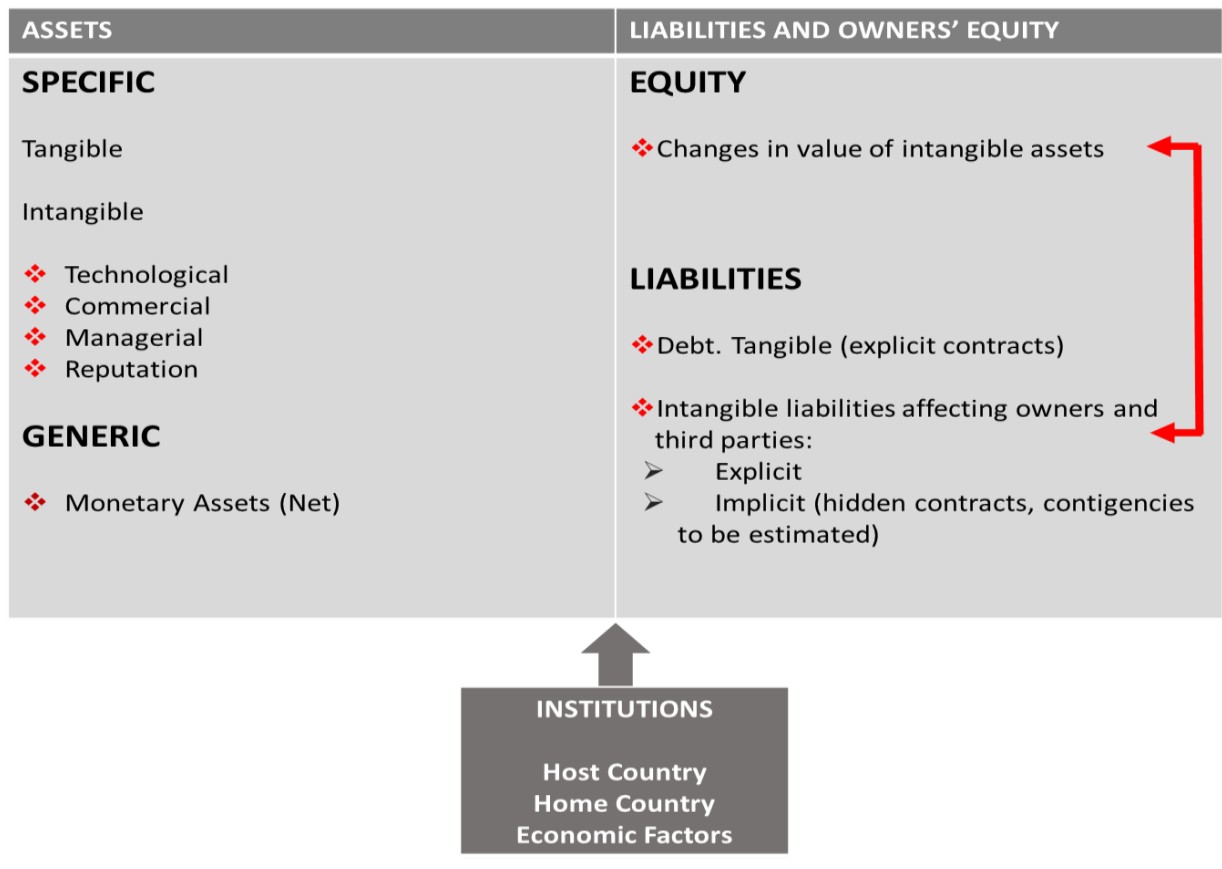

**Figure 1.** Accounting equation balance.

If the company carries out a good management of its intangible assets, it contributes to increase its value and, therefore, there is an improvement of its competitiveness and reputation. In the same sense, a decrease on intangible assets means a decrease on the value of the firm, the appearance of intangible liabilities and consequently, the effect of them on reducing its competitiveness and reputation.

Not all the intangible liabilities sources have the same consideration. Although it is true that some intangible liabilities may arise suddenly, such as when a catastrophic event happens, it is also true that the company, as far as it may know its probable risks, has the potential to mitigate the impact of this type of events on its intangible assets through, for example, hiring insurance. Following this discussion, a lack of assurance may involve the appearance of intangible liabilities arising on mismanagement.

For those firms that are not listed in capital markets, it is necessary to estimate its fair market value and compare it with the value of assets to estimate the value of intangible assets. There are several methods of intellectual capital measurement and valuation but going deeper on this matter is not the proposal of this paper [39]. However, assessing the risks arising from these intangible liabilities is becoming increasingly necessary given that they determine access to financing. These aspects are some of those that are included as elements to be considered in the taxonomy of sustainable financing on which Europe has been working since 2018, and whose fundamental objective is to meet the "EU's climate and energy targets for 2030 and reach the objectives of the European green deal through redirecting money towards sustainable projects that make economies, businesses and societies—in particular, health systems—, more resilient against climate and environmental shocks" [11].

## 5. Source of Intangible Liabilities

When hidden liabilities emergence and contingencies are recorded to support the future payments to third parties (fines, compensations, v.gr.), can be expected a loss of value of intangible assets though out a loss of reputation. Loss of value of the specific assets can be interpreted as a loss of firm's competitiveness. The decrease on assets value cannot be only reflected on the decrease in the equity of the firm, but it also can be reflected on its liabilities, increasing them at the time that these loses were accompanied by payments obligations to third parties. This fact also is expected to be correlated to a loss of firm's reputation.

Ethical behaviour, as a source of firm reputation, is valuable for capital markets [43–45]. Firms with a high level of consumer satisfaction, that have a positive relationship with corporate responsibility (CR), can generate more cash flows and reduce its future volatility, which will end up as a greater market value of the firm [46–48]. Additionally, a direct relationship has been found between high standards of CR performance and high market capitalization [49,50].

Expectations based on wrong ideas may lead to weak strategies being brought sooner or later to the recognition of intangible liabilities. The implicit options and consequences of investment decisions need to be analyzed. The implementation of a bad idea and inappropriate decisions or a wrong strategy can be considered as intangible assets depreciation [51]. At the same time, if this depreciation implies obligations, firm's intangible liabilities arise. Decreases in the value of intangible assets will have a reflection on financing sources, reducing the value of equity. It can also be the case that in addition to producing a reduction in equity, intangible liabilities appear (for example, when a firm has the obligation to repair damages caused by irresponsible behavior).

However, not all intangible liabilities have their origin in intangible assets depreciation. Sources of Intangible liabilities may arise from three sources:

(a)  Contingencies. From a broader perspective, intangible liabilities may be connected to contingencies that bring about a reduction in competitiveness of the firm and consequently a reduction in firm equity value and reputation. The firm's loss of competitive advantages (a loss of value of economic capital) is expected to reduce its equity value. To anticipate this outcome, the company may have to assume intangible liabilities (possible obligations and investment payments) to maintain the value of intangible assets. In certain cases, the amount or the maturity of the liability may be undetermined. Contingencies obligations, current and likely future obligations of the company, because of past actions, must be recognized in the balance sheet

as a reserve fund. Although not all liabilities are bound to occur, the likelihood of them occurring needs to be considered. Accepting and recognizing the existence of intangible liabilities may be a way of avoiding the depreciation of intangible assets and, at the same time, to preserve the firm reputation.

(b) Breach of contracts terms that may arise from future payment obligations.

(c) Risky decision such as hidden bad practices adopted by the firms, expecting not to be discovered by the stakeholders, which may imply future intangible liabilities. There are examples of risky decisions based on bad quality of products (product adulterations and products of inferior quality), danger for health (customer harmed), litigation against companies (antitrust violation), risky working conditions (factory disaster in LDCs, for example in Bangladesh), corruption, environmental intangible obligations (environmental cleanup), inadequate (inefficient) geographical location of plants, not abiding by standards of developed countries but rather adapting the low quality of local standards. Problems of avoiding relief for the victims of factory disasters in less developed countries, such as, in short term emergency assistance, proper medical care, compensation and assistance to the injured workers, compensation to families for the loss of a bread-winner, and the long run, designing programs to "empower" those most vulnerable people (i.e., Widows and their Children of those deceased in these accidents) to free enjoy the mentioned compensations in their communities of residence [48]. Accidents, irresponsibility, and negligence are sources of intangible liabilities. Companies are aware of their misbehavior given that, despite the possibility of communicating the various risks they incur or may incur through the various reports of available non-financial information (Global Reporting Initiative, Accountability, ISO26000, among others), prefer to hide their bad practices for the purpose of not being discovered and thus maintaining their reputation.

*5.1. Some Real Cases*

In the primary sectors, cases of oil and residual spills into the oceans and rivers have been common such as for example: Exxon Valdez in Alaska (1989), Baia Mare gold mine in Rumania (2000), gold mine in Papua Nueva Guinea, Gulf of Mexico by BP (2010), Bophal, India (1984), essentially due to obsolete equipment and installations and the use of untrained personnel; the use of child labor in India´s supply chains (for MNEs such as Bayer, Advanta, Monsanto, or Emergent Genetic), Vietnam (Nike), use of labor forced in Burma (Total). These are all cases of negative advertising (and bad reputation), all of which the public were informed of by ONGs, and all of them had a great impact on their stakeholders, because of globalization and the possibility given by the social media of finding out any information in real time. Additionally, in the same categories we can cite Chiquita Brands, putting workers in Latin America in danger by forcing them to utilize pesticides [52]. Coca Cola had to pay important sums in India for the use of water with pesticides [53]. Damage by Shell in Niger, ChevronTexaco in Ecuador, ExxonMobil and Petrobas in Chad and Cameroun [54] was followed also by the payment of large sums of money.

In financial area, bad practices and manipulation of interest rates and foreign exchange rates by banks have affected references indexes (EURIBOR, LIBOR, TIBOR) and to the international financial crisis creating lack of confidence, loss of reputation and intangible liabilities both for the banks and for the society as such (recapitalization of capital of banks, unemployment, loss of value of assets). The main banks in USA were penalized with thousands of millions of dollars for bad practices. Among these banks and besides bad practices with minority clients, Deutsche bank were penalized for manipulate the interbank market and foreign exchange rates; these facts together with bad results, made share prices were down about 90%. Deutsche bank registered a reserve fund of EUR 5000 million to paid legal procedures. Wells Fargo has been penalized with USD 185 million for having invented about two million bogus consumer accounts. In Europe fines to banks of about USD 485 million to each of the following: JP Morgan (US), Credit Agricole (France), HSBC

(UK). In 2013 after cooperation with EU Authorities, 830 million fines to Deutsche Bank, Barclays, Royal Bank of Scotland and Societe Generale.

The Mexican Financial Regulator (Comisión Nacional Bancaria y de Valores, CNBV) has imposed a fine (March 2016) on the Spanish Multinational OHL of EUR 3.6 million (71.7 million pesos) for a corruption scandal due to inflated costs in the project of the State of Mexico. Executives have also been fined with EUR 511,000. Several corruption scandals, uncompleted contracts, plus management crisis of this MNE, accompanied by poor financial results end up with a junk bonds calcification of the OHL debt by Moody´s. During the first semester of 2016 the OHL share fall 60%. One week later the announcement of two new projects to be developed by the American subsidiary of OHL (OHL USA and Judian) was in parallel with the confirmation of Fitch of BB rating of OHL (three levels above the one decided by Moody´s). The Investment Fund Tyrus in august 2015 bought the OHL share at EUR 14,8. In august 2016 the price was at less than EUR 2; Tyrus demanded compensation for those losses (perhaps considered that was hidden intangible liabilities in 2015).

In 2015, The Transport and Environment Organization published a Report entitled "The Dirty Thirty" about the 30 dirty car models which included many well-known car models, belonging to Volkswagen. Some of these models used up to ten times the maximum $NO_2$ allowed. The report said that firms has developed more innovating techniques to hide the emissions than to reduce them. Volkswagen was the first firm to be uncovered and had to make a reserve fund of EUR 16,200 million, besides facing the strong negative effect on its reputation and the decrease in share price of more than 17% in one day. Volkswagen provided a reserve fund of USD 18,000 million to cover the legal charges and clients' compensations. In USA plus an estimated fine of USD 1200 million. Volkswagen´s Shareholders (1400 legal) demanded EUR 8200 million as compensation.

An example of breaching of contract terms is the behavior of Spanish Multinational Sacyr, building the Panama Canal. This company requested a cost overrun of USD 2390 million. The main causes for it were quality, unexpected faults, strikes and regulatory changes. Panama Canal Authority (ACP) rejected the claims of the Consortium. The Miami Court of Arbitration is not expected to solve the conflict before 2020 and the outcome is uncertain. Moreover, this intangible liability the construction of the Panama Canal created a positive reputation for Sacyr. Most likely, the net impact is a positive one and implies a net increase in intangible assets.

Apple is the world's largest company and at the same time, following the index elaborated by Fortune, the most admired company in the world. However, the European Union has encouraged Apple to pay a fine of EUR 13bn originating from Ireland's tax arrangements with Apple between 1991 and 2015, which allowed the US company the following: on one hand, it allowed Apple to attribute sales to a "head office" that only existed on paper and could not have generated such profits. On the other hand, the deal allowed Apple to pay a maximum tax rate of just 1%. In 2014, the tech firm paid tax at just 0.005% being the usual rate of corporation tax in Ireland is 12.5%. The result was that Apple avoided tax on almost all the profit generated from its EUR multi-billion sales of iPhones and other products across the EU's single market. It booked the profits in Ireland rather than the country in which the product was sold.

### 5.2. Accounting Proposal of Intangible Liabilities

Under the proposed conceptual approach, we consider the possibility for the firm to recognize its intangible liabilities. IAS 37 for "Provisions and Contingent Liabilities and Contingent Assets" defines provisions as liabilities for which there is uncertainty about timing or amount and indicates that a provision is recognized when: (a) the company has a present obligation as a result of a past event; (b) it is likely that the company will have to expend resources embodying economic benefits to settle the obligation; and (c) also it can be estimated reliably the amount of the debt.

This standard also defines a contingent liability as: (a) any possible obligation that arises from past events and whose existence will be confirmed only if they occur, or else if they do not occur, one or more future uncertain events are not entirely under the control of the company; or (b) a present obligation that arises from past events but is not recognized in the financial statements. In any case, the possibility of the provision or the contingent liability arising should be greater than the possibility that it does not arise to be recognized in accounting. However, regardless of whether it broke out the contingency or not, the company must report on the obligation in the financial statements, excepting if an outflow of resources embodying economic benefits is remote.

Accounting for contingent liabilities is one of the most critical areas of financial accounting, mainly because of the high degree of subjectivity involved in the recognition and valuation of uncertain situations [55]. The point here is to decide when the contingency emerges, due to the management of the company has room for maneuver in the recognition and value of the facts, being able to move the profits or losses from one year to another under legal cover [56–58]. To use that room of maneuver, represents an enormous attraction for the executives of the companies, especially in times of crisis.

## 6. Results and Discussion

The developed analysis contrasts the following hypothesis:

**Hypothesis 1 (H1):** *Factors impacting more on perception of reputation by stakeholders are inked to the management style of the MNE.*

**Hypothesis 2 (H2):** *Factors such as innovation, human resources management, corporate social responsibility, quality of products and/or services and global competitiveness are consequence of management decisions.*

**Hypothesis 3 (H3):** *Factors as innovation and quality of products and services are relevant in the study of reputation variation.*

The obtained results analyzed using the constructors (Table 1) are the following:

**Table 1.** Factor analyses.

| Dimension. Constructors | Code | Factor Analysis Load | Internal Consistency and Reliability Statistics |
|---|---|---|---|
| Reputation | Use of corporate assets | 0.879 | Alpha Cronbach: 0.888 Composite reliability: 0.923 AVE: 0.750 |
| | Quality of management | 0.866 | |
| | Financial Soundness | 0.774 | |
| | LT Investment Value | 0.937 | |
| Financial Performance 1 | Assets | 0.984 | Alpha Cronbach: 0.816 Composite reliability: 0.914 AVE: 0.728 |
| | Liabilities | 0.963 | |
| | Shareholders' Equity | 0.869 | |
| Financial Performance 2 | Capitalization | 0.783 | Alpha Cronbach: 0.780 Composite reliability: 0.958 AVE: 0.884 |
| | Revenue | 0.737 | |
| | Book Value | 0.942 | |
| | Tangible assets | 0.931 | |

The first constructor is composed by corporate assets, quality of management, financial soundness and long-term investment value and shows that the factors impacting the most on perception of reputation by stakeholders. This analysis shows that the factors impacting more on perception of reputation by stakeholders are those linked to the management style of the MNE, the remaining factors being (innovation, human resources management, corporate social responsibility, quality of products and/or services and global competitiveness) consequences of management decisions. This result validates H1, what implies that the

management style really impacts on reputation perceived by stakeholders. This is aligned with other studies on intangible liabilities [14]

The second constructor is formed by two variables: the first financial synthetic variable constructed through the factor relationship among assets, liabilities and shareholders' equity, coming from the basic accounting equation and the relationship between economic resources and financing sources The second synthetic financial one is constructed through the factor relationship of market capitalization, revenue, book value and tangible assets, as it is pointed out in Ryan [13]. This validates H2, due to the fact that those variables are a consequence of management decisions. This is partially aligned with the results included in [13–16,41,58].

The third constructor is formed by two variables: the first financial synthetic variable constructed through the factor relationship among assets, liabilities and shareholders' equity, coming from the basic accounting equation and the relationship between economic resources and financing sources and the relationship between Innovation and Quality of Products and Services. This constructor is discarded because it seemed irrelevant when studying reputation variation, which invalidates H3.

In all cases, values are higher than the thresholds defined in the literature, 0.7 for Cronbach's alpha and 0.5 for AVE. The discriminant validity between constructors is also analyzed and shows that the correlations are lower than the square root of AVE.

Once the constructors were determined, we studied the correlation for each valid dimension and therefore, the correlation between the variation of the reputation and financial constructors defined, with the results that are shown below (Table 2):

**Table 2.** Correlations between constructors derived from dimensions.

|  |  | Reputation | Financial Performance 2 | Financial Performance 1 |
|---|---|---|---|---|
| Reputation | PearsonCorrelation | 1 | 0.156 | 0.087 |
|  | Sig. (bilateral) |  | 0.307 | 0.571 |
|  | N | 46 | 45 | 45 |
| Financial performance 2 | PearsonCorrelation | 0.156 | 1 | 0.693 ** |
|  | Sig. (bilateral) | 0.307 |  | 0.000 |
|  | N | 45 | 49 | 49 |
| Financial performance 1 | PearsonCorrelation | 0.087 | 0.693 ** | 1 |
|  | Sig. (bilateral) | 0.571 | 0.000 |  |
|  | N | 45 | 49 | 49 |

** The correlation is significant at the 0.01 level (bilateral).

The analysis carried out shows that there is no correlation between the change in the reputation and financial impact contained in the constructors. However, there is a logical positive correlation between the generated synthetic financial variables (H1). Therefore, there is no correlation between a perceived loss of reputation of a company and loss of value in certain financial variables included or not in that loss of reputation. Although there is not a positive correlation between the synthetic variable "reputation", a variable that collected those factors explaining the decrease in reputation, and the two synthetic financial variables (Table 3).

It shows how once we consider individually each reputational factor they generate a positive correlation in relation to all individual financial variables, which explains the effect of reputation on the financial variables of the MNE and, therefore, the management style of the MNE, so hypothesis H1 is confirmed what is aligned with other studies related to intangible liabilities and their effects on their recognition and effects for the companies [13,42,56,58]. As it has been described, much of these intangible liabilities or assets have a social or environmental origin; therefore, acting on sustainability would imply a change in reputation perception.

**Table 3.** Independent correlations among all the variables studied.

| | | Market Capitalization | Revenue | Assets | Book Value | Tangible Assets | Liabilities | Shareholders' Equity |
|---|---|---|---|---|---|---|---|---|
| S'Use corporate assets | PearsonCorrelation | 0.097 | −0.130 | 0.063 | 0.167 | 0.159 | 0.051 | 0.162 |
| | Sig. (bilateral) | 0.521 | 0.393 | 0.683 | 0.272 | 0.298 | 0.740 | 0.288 |
| | N | 46 | 45 | 45 | 45 | 45 | 45 | 45 |
| Quality of management | PearsonCorrelation | 0.021 | −0.180 | −0.024 | 0.056 | 0.153 | −0.030 | 0.050 |
| | Sig. (bilateral) | 0.892 | 0.238 | 0.877 | 0.715 | 0.315 | 0.844 | 0.746 |
| | N | 46 | 45 | 45 | 45 | 45 | 45 | 45 |
| Financial Soundness | PearsonCorrelation | 0.245 | 0.089 | 0.118 | 0.300 * | 0.248 | 0.091 | 0.289 |
| | Sig. (bilateral) | 0.101 | 0.560 | 0.442 | 0.045 | 0.101 | 0.550 | 0.054 |
| | N | 46 | 45 | 45 | 45 | 45 | 45 | 45 |
| Long-Term Investment Value | PearsonCorrelation | 0.125 | 0.043 | 0.021 | 0.150 | 0.198 | 0.007 | 0.141 |
| | Sig. (bilateral) | 0.408 | 0.778 | 0.892 | 0.325 | 0.193 | 0.964 | 0.356 |
| | N | 46 | 45 | 45 | 45 | 45 | 45 | 45 |

* The correlation is significant at the 0.01 level (bilateral).

## 7. Conclusions

The main contribution of this paper is the definition of Intangible Liabilities as a new or not deeply studied concept in International Business research. We identify the sources of this implicit intangible liability that it may be related to the competitiveness of the firm. We also argue the correlation expected between the competitiveness of the firm and the specific and singular intangible asset such as reputation.

This paper also contributes to underlying the relevance of firms' reputations and their connection to ownership advantages of the firm and, also in certain cases, with intangible liabilities. An efficient use of economic capital will be reflected in the value of the firm and in another singular and important intangible asset: its reputation. In this respect, a loss or gain of reputation should be correlated to the competitiveness of the firm. A reduction in the reputation might be due to bad practices, bad decisions that may lead to intangible liabilities with the obligation of future payments to third parties. Thus, the depreciation of intangible assets has an influence on a reduction in equity value and, in certain cases due to contingencies, possibly will emergence intangible liabilities.

Contingencies and not anticipated events (not hedged neither insured) as well as hidden liabilities and bad practices may be sources of intangible liabilities, correlated to payment obligations to third parties. The potential intangible liabilities are implicit and when they are demanded by external actors (stakeholders) become intangible liabilities that will have to be shown in the financial accounts and end up with monetary payments to third parties. Anticipated contingencies may be reflected in the provision of reserve funds. Implicit (Intangible) liabilities once they become explicit (known) payments obligations to third parties must be registered in the financial accounts of the firm. Most of this intangible would come from sustainable actions under a social and environmental point of view, so act on them should boost reputation.

In the empirical analysis, we have taken reputation as a constructor to indicate the potential of generating implicit intangible liabilities. We conclude that reputation is related to the quality and style of management, long term investment and financial soundness and not only to financial performance. Our analysis shows that the factors (corporate assets, quality of management, financial soundness, and long-term investment) with a highest impact on perception of reputation by stakeholders are those linked to the management style of the MNE. The remaining factors as innovation, human resources management, cor-

porate social responsibility, quality of products and/or services and global competitiveness, are consequences of management decisions.

The limitation of the research was the small number of companies included in the sample and the period considered, which is prior to the definition of the European sustainable financing taxonomy. Both limitations offer a base line for future research, through broadening the data base, including not only more companies and periods, but other factors that allow us to explore new relationships such as reputation and liabilities of foreignness.

This research provides a significant practical contribution to MNCs management because they could better manage their intangible, assets and liabilities, in order to increase financing access, decreasing risks and improve reputation through the inclusion of sustainability in their policies and strategical performance.

**Author Contributions:** Conceptualization, M.-J.G.-L. and J.-J.D.; methodology, M.-J.G.-L. and J.-J.D.; validation, M.-J.G.-L., C.A.-P. and J.-J.D.; formal analysis, M.-J.G.-L.; investigation, M.-J.G.-L.; resources, M.-J.G.-L.; data curation, M.-J.G.-L.; writing—original draft preparation, M.-J.G.-L. and C.A.-P.; writing—review and editing, M.-J.G.-L. and C.A.-P.; supervision, J.-J.D.; project administration, M.-J.G.-L.; funding acquisition, C.A.-P. All authors have read and agreed to the published version of the manuscript.

**Funding:** Action financed by the Community of Madrid within the framework of the Multi-year Agreement with the Universidad Politécnica de Madrid in the line of action Program of Excellence for University Teaching Staff- Echegaray Grant.

**Institutional Review Board Statement:** Not applicable.

**Informed Consent Statement:** Not applicable.

**Data Availability Statement:** Not applicable.

**Conflicts of Interest:** The authors declare no conflict of interest.

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
