# Peer review of "Managing Reputation in MNEs through Intangible Liabilities"

_sustainability, doi:10.3390/su14053041_

Round 1
Reviewer 1 Report
Review of literature on intangible liabilities and reputation has not been conducted well. This is because the authors haven't included in more recent papers on this specific field of research. The autors should do more review of more current literature and add it to the discussion in this paper. I would see the following changes:
- A more detailed analysis of the existing research gaps.
- A more detailed analysis of the existing literature on the subject area. Aditionally, the autors should elaborate more about what academic contributions have been made by this paper.
Why 2017, 2018 and 2019 are chosen for the research, when it is already 2022?
The discussion part should present a direct comparison of the results to previous studies.
The methodology: This section should also identify strengths and weaknesses of the methodology.
Author Response
Dear Reviewer 1
We would like to thank your patience and great comments and suggestions which much improve our research.
Please, consider our notes. We hope to have done our best to take into consideration your comments.
Best regards

Reviewer 2 Report
The paper concentrates on the important issue from the perspective of the enterprises (especially the multinational firms) in regard to the intangible liabilities and their effect on the value of the firm and the firm reputation.
However, before publication the paper needs following improvements:
- Motivation is weak. Authors could better define the aim of the paper. Moreover, they could better point out the existing gap in the literature.
- Introduction should be extended, i.e. brief information on used methods, why this method was chosen instead of etc.
- Literature review and improvements in regard to literature background should be made. Examples of articles that may be useful for the authors identifying similar contributions and better show what is the specific contribution.
- de Almeida Aguiar, Gabriel; Peres Tortoli, Júlia; Pinto Figari, Anelise Krauspenhar; Pimenta Junior, Tabajara. (2021) Analysis of the influence of intangible assets on the performance of Brazilian companies. Brazilian Journal of Management / Revista de Administração da UFSM, 14(4), 907-931. doi: 10.5902/1983465944075
- Lewandowska, A. (2021) Interactions between investments in innovation and SME competitiveness in peripheral regions. Polish case study. Journal of International Studies, 14(1), 285-307. doi: 10.14254/2071-8330.2021/14-1/20
- Bryl, Łukasz (2020) Intangible assets in the process of internationalization. International Journal of Management & Economics, 56(1), 63-78. doi: 10.2478/ijme-2020-0004
- Khan, Sher Zaman; Yang, Qing; Waheed, Abdul (2019) Investment in intangible resources and capabilities spurs sustainable competitive advantage and firm performance. Corporate Social Responsibility & Environmental Management, 26(2), 285-295. doi: 10.1002/csr.1678
In my opinion these four references cannot be omitted.
- The results are well-analysed and described, but I miss the Discussion section. The results are presented clearly, but the findings are not compared and contrasted with relevant literature. Linking theoretical considerations with empirical findings and providing some planning insights is critical in a journal with the scope of Sustainability.
I hope that my comments are helpful to you as you continue your work on this project. Good luck!
Author Response
Dear Reviewer 2
We would like to thank your patience and great comments and suggestions which much improve our research.
Please, consider our notes. We hope to have done our best to take into consideration your comments.
Best regards

Reviewer 3 Report
The title of the article is very unfortunate. The title of the article should hint at the main problem which the authors considered in the paper.
Why is this research important to the reader? What kind of gaps does it cover?
The article submitted to «Sustainability» journal, but it not reflected sustainability.
The abstract is weak. Kindly enhance it by including novelty and need of the study.
Expand MNE in introduction and then start using abbreviation.
Introduction should be enhanced by including significance of intangible assets and reputation for business sustainability.
Include research questions and hypotheses.
More justification should be provided for the choice of factors in the section Research model and methodology.
Line 196 – 32. Check references.
Add more resources.
Line 337 - UNCTAD, 2008. Add resource.
The result and discussion section are very much disorganized. The authors should clearly state which studies are in line with their findings and which contradicts.
Provide managerial, theoretical and research implications.
Enhance the conclusion section in line with the research questions.
Include more limitations and future research directions.
Author Response
Dear Reviewer 3
We would like to thank your patience and great comments and suggestions which much improve our research.
Please, consider our notes. We hope to have done our best to take into consideration your comments.
Best regards

Round 2
Reviewer 3 Report
The authors of the article took into account our earlier recommendations. Thanks.